# Automated evaluation of colon capsule endoscopic severity of ulcerative colitis using ResNet50

**Naoki Higuchi**[1], **Hiroto Hiraga**[1]\*, **Yoshihiro Sasaki**[2], **Noriko Hiraga**[1], **Shohei Igarashi**[1], **Keisuke Hasui**[1], **Kohei Ogasawara**[1], **Takato Maeda**[1], **Yasuhisa Murai**[1], **Tetsuya Tatsuta**[1], **Hidezumi Kikuchi**[1], **Daisuke Chinda**[1], **Tatsuya Mikami**[1], **Masashi Matsuzaka**[2], **Hirotake Sakuraba**[1], **Shinsaku Fukuda**[1]

**1** Department of Gastroenterology and Hematology, Hirosaki University Graduate School of Medicine, Hirosaki, Japan, **2** Department of Medical Informatics, Hirosaki University Hospital, Hirosaki, Japan

\* hhiraga@hirosaki-u.ac.jp

**Data Availability Statement:** All relevant data are within the article and its Supporting Information files.

## Abstract

Capsule endoscopy has been widely used as a non-invasive diagnostic tool for small or large intestinal lesions. In recent years, automated lesion detection systems using machine learning have been devised. This study aimed to develop an automated system for capsule endoscopic severity in patients with ulcerative colitis along the entire length of the colon using ResNet50. Capsule endoscopy videos from patients with ulcerative colitis were collected prospectively. Each single examination video file was partitioned into four segments: the cecum and ascending colon, transverse colon, descending and sigmoid colon, and rectum. Fifty still pictures (576 × 576 pixels) were extracted from each partitioned video. A patch (128 × 128 pixels) was trimmed from the still picture at every 32-pixel-strides. A total of 739,021 patch images were manually classified into six categories: 0) Mayo endoscopic subscore (MES) 0, 1) MES1, 2) MES2, 3) MES3, 4) inadequate quality for evaluation, and 5) ileal mucosa. ResNet50, a deep learning framework, was trained using 483,644 datasets and validated using 255,377 independent datasets. In total, 31 capsule endoscopy videos from 22 patients were collected. The accuracy rates of the training and validation datasets were 0.992 and 0.973, respectively. An automated evaluation system for the capsule endoscopic severity of ulcerative colitis was developed. This could be a useful tool for assessing topographic disease activity, thus decreasing the burden of image interpretation on endoscopists.

## Introduction

Ulcerative colitis (UC) is an idiopathic, diffuse, and chronic inflammatory disease of the colonic mucosa [1]. A therapeutic target for UC has been reported as endoscopic mucosal healing or the Mayo endoscopic subscore (MES) of 1 [2]. To objectively evaluate the severity of UC, we previously characterized its endoscopic features, including mucosal patterns (spatial arrangements of mucosal color) and the degree of roughness on the mucosal surface [3–5].

**Funding:** The authors received no specific funding for this work.

**Competing interests:** The authors have declared that no competing interests exist.

Table 1. Mayo endoscopic subscore (MES).

| Grade | Endoscopic findings |
|---|---|
| 0 | No friability and granularity and intact vascular pattern. |
| 1 | Mild erythema or decreased vascular pattern. |
| 2 | Marked erythema, absent vascular pattern, friability, and erosions. |
| 3 | Spontaneous bleeding and ulceration. |

For several years, convolutional neural networks (CNNs), a deep learning method, have facilitated automated evaluation of colonoscopic severity or the MES [6] (Table 1) in patients with UC [7–9]. Becker et al. presented a CNN-based grading algorithm for colonoscopy videos of patients with UC [10]. Although colonoscopy is the gold standard modality for disease severity and extent, the low acceptability of colonoscopy must be considered [11]. Wireless capsule endoscopy (WCE) is an established diagnostic tool for the evaluation of various small bowel abnormalities [12], such as bleeding, mucosal pathology, and small bowel polyps. Colon capsule endoscopy (CCE) was developed in 2006 to allow non-invasive visualization of the colon [13]. Newer CCE-2 devices, such as the PillCam COLON 2 (Medtronic, Dublin, Ireland), have facilitated imaging that is superior to that of the first-generation CCE devices. Hosoe et al. [14] reported a high correlation (p = 0.797) between the Matts endoscopic score [15] determined using CCE-2 images and conventional colonoscopy. In the field of WCE, a CNN-based diagnostic program was challenged to recognize celiac disease [16], hookworm infection [17], and small intestine motility characterization [18]. CNN-based computer-aided diagnosis (CAD) would help reduce reading time, oversight, and burden on physicians by automatically detecting gastrointestinal tract abnormalities. Thus far, several computer-aided methods have been investigated for reading capsule endoscopy images [19–23]. A major limitation of a CNN-based diagnostic program using WCE is difficult to develop because CE image quality is usually poor due to hardware and light limitations and low resolution (320 × 320 pixels). Additionally, WCE image quality is further limited by various orientations because of the free motion of the capsule and various extraneous matters, such as bile, bubble, food, and fecal material.

To the best of our knowledge, there are no studies on the automated evaluation of capsule endoscopic severity in patients with UC. In addition, a single MES has been assigned to a single endoscopic image [7–9], whereas MES is often different from each region in a single endoscopic image, especially during the resolution phase in patients with UC. This study aimed to develop a CAD system for evaluating the spatial ratio of severities in a single endoscopic image along the entire length of the colon, yielding a topographic map of severity in routine CCE-2 examinations for patients with UC. Hence, the burden of image interpretation by endoscopists would be reduced.

## Methods

### Preparation of endoscopic images

A CCE-2 device's (PillCam® COLON2) video files (MPEG files with a size of 576 × 576 pixels) were obtained from our hospital (Hirosaki University Hospital, Aomori, Japan) and used for this single-center study. There were 31 video files from 22 patients with UC (24 moderate disease and 7 mild disease) who underwent CCE-2 between March 7, 2018, and September 2, 2020. A MOVIPREP and caster-oil regimen was adopted for pre-treatment [24]. The CCE-2 device has two cameras with a 172˚ angle of view on both ends (forward and backward), capturing images at 4 or 35 frames per second depending on its moving speed [25]. Therefore, a

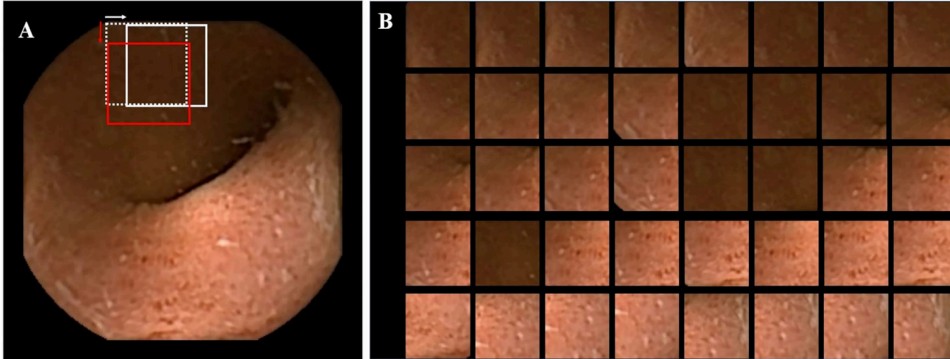

**Fig 1. Preparation of dataset.** A, An extracted still picture with the size of 576 × 576 pixels. The patch dataset images (128 × 128 pixels) were trimmed from the still picture starting from the left upper corner (white dotted patch), rightwards (white solid patch), then downwards (red solid patch) at every 32-pixel-strides (white and red arrows) over the entire effective region of the still picture. B, A total of 40 patches eligible for analysis.

single recording session comprised a set of two video files. The raw data were automatically edited and converted to digest video files with a fixed frame rate of 25 frames per second and a playback length from 2.3 to 30.4 min using RAPID® v8.3 (Medtronic, Dublin, Ireland). With reference to the location profile of the pill, the pair of examination video files was manually partitioned into four segments: the cecum and ascending colon, transverse colon, descending and sigmoid colon, and rectum. Fifty still pictures (jpg files with a size of 576 × 576 pixels) were extracted from each partitioned video. To evaluate regional MES, a patch (128 × 128 pixels) was trimmed from the effective region of the still picture at every 32-pixel-stride (Fig 1A). Patches with blackouts or higher-intensity areas were automatically excluded from the analysis. Blackout patch was defined as that where pixel counts with low intensity (<70) exceed 1% of the total pixels (128 × 128) in the red frame, and higher-intensity patch was considered when pixel counts with high intensity (>230) exceed 5% of the total pixels in the red frame. In case of this still picture, a total of 40 patches for analysis were extracted (Fig 1B). With reference to the original still picture, five well trained endoscopists scored and classified these trimmed patch images into six categories: 0) MES0, 1) MES1, 2) MES2, 3) MES3, 4) inadequate quality for evaluation, and 5) ileal mucosa. A total of 739,021 patch images eligible for analysis were classified. Finally, two authors (HH and HN) reviewed and confirmed the endoscopic classifications. In the present study, white scars and inflammatory polyps were classified as MES0 because they are inactive findings from the viewpoint of disease severity. Representative images of the six categories are shown in Fig 2. This study was approved by the ethics committee of Hirosaki University Graduate School of Medicine on July 4, 2017 (approval number: 2017–1046). We obtained the informed consent from all patients in writing, prior to participation of this study.

## Inclusion and exclusion criteria

We excluded UC patients with severe conditions from this study, because the MOVIPREP regimen would have been intolerable for them, and additionally it would be ethically wrong to cause harm. No inclusion or exclusion criteria were specified for image classification by endoscopists. Trimmed patch images with blackouts or higher-intensity areas were automatically excluded before classification. This study aimed to establish an effective severity classification that can be used in any common clinical condition without human intervention.

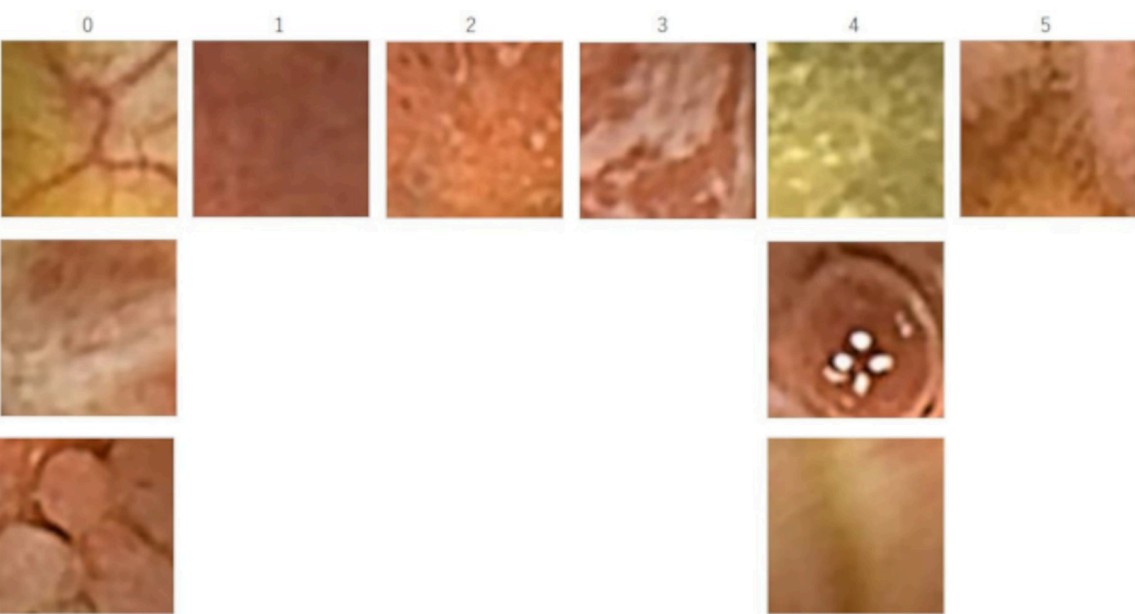

**Fig 2.** Representative images of the six categories: 0) MES0, normal (upper), white scar (middle) and inflammatory polyps (lower); 1) MES1, decreased vascular pattern; 2) MES2, absent vascular pattern, friability, and erosions; 3) MES3, ulceration; 4) inadequate quality for evaluation, effluent with residue (upper), bubble (middle) and motion blur (lower); and 5) ileal mucosa.

## Training and validation dataset

The training dataset comprised 483,644 images from 15 patients with UC who underwent CCE-2 from March 7, 2018, to January 20, 2020. All images were manually classified into six categories, as mentioned above.

To assess the performance of the proposed CNN as a severity classifier, another set of 255,377 images from eight patients with UC who underwent CCE-2 from January 21, 2020, to September 2, 2020 was used. These images were manually classified into six categories to validate the accuracy of CNN using the same method. The training and validation datasets are listed in Table 2.

## Architecture of CNN

ResNet50 (a CNN) and Pytorch (a moving framework) were utilized [26]. ResNet50 without pretraining was imported from the Pytorch library (Torchvision. models). The original patch images with $128 \times 128$ pixels were converted into images with $224 \times 224$ pixels. We tuned the

**Table 2. Number of images in each training and validation dataset.**

| No | Categories | Training data set | Validation data set |
|---|---|---|---|
| | | Number of pictures | Number of pictures |
| 0 | MES0 | 112544 | 109721 |
| 1 | MES1 | 60367 | 14753 |
| 2 | MES2 | 8710 | 3507 |
| 3 | MES3 | 21153 | 20736 |
| 4 | Inadequate quality for evaluation | 280368 | 106577 |
| 5 | Ileal mucosa | 502 | 83 |
| | Total | 483644 | 255377 |

hyperparameters, which were set by a human, as follows: optimizer, Adam; loss function, cross-entropy loss; number of training epochs, 50; batch size, 256; learning rate, 0.00025 via trial and error; and number of outer layers, six classes.

## Severity of a single still picture

Although UCEIS (Ulcerative Colitis Endoscopic Index of Severity) is recognized to be a more accurate assessment of mucosal severity for patients with UC as compared to MES, UCEIS is designed to evaluate the severity with only a single image. In contrast, capsule endoscopy has an advantage in that it obtains and evaluates serial images of the whole colon. We therefore selected MES and not UCEIS to evaluate the severity and to construct a topographic map of the severity. Four examples of still pictures before and after the automated classification are shown in Fig 3. The patch images trimmed from the still picture were classified into six classes using trained ResNet50. The patches classified into MES0, MES1, MES2, and MES3 are illustrated by right gray, yellow, magenta, and red open patches, respectively, in the second to fifth columns of Fig 3. When the patches classified into MES0 are numbered as $S_1$, $S_2$,. . ., $S_n$, area0 (area of MES0) were defined by the union of a collection of all elements i.e., area0 = $S_1 \cup S_2 \cup . . . \cup S_n$. The areas for MES1, MES2, and MES3 were similarly given by area1, area2, and area3, respectively. Provided total area = area0+ area1+ area2+ area3, endoscopic severity

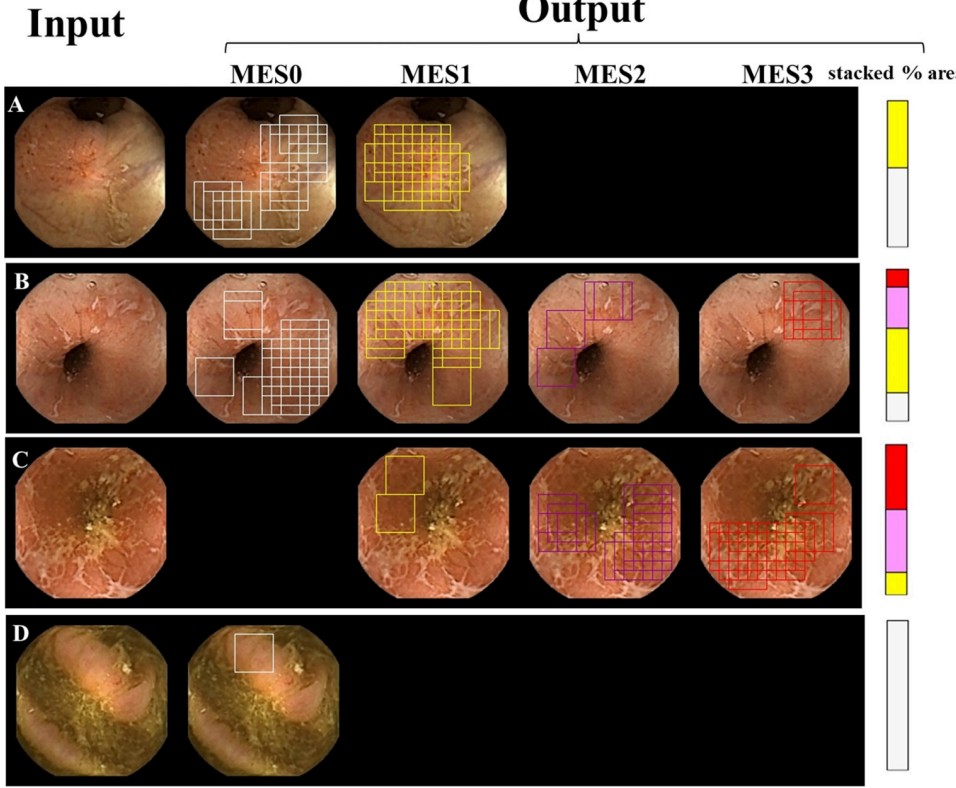

**Fig 3. Algorithm for evaluating the severity of a single still picture.** Patches trimmed from input images (left columns of A–D) were classified into MES0 (dark gray open square), MES1 (yellow open patch), MES2 (magenta open patch), and MES3 (red open patch). Area0 (area of MES0) is defined by the union of the dark gray open patches. Similarly, area1, area2, and area3 by that of yellow, magenta, and red open patches, respectively. Severity is expressed by the stacked bar graph, composed of % area: white, MES0; yellow, MES1; magenta, MES2; and red, MES3 (right columns of A–D).

was expressed by the stacked bar graph composed of % areas, including area0/total area × 100 (light gray part), area1/total area × 100 (yellow part), area2/total area × 100 (magenta part), and area3/total area × 100 (red part) in the right column of Fig 3. A serial stacked bar graph was automatically created along the entire length of the colon, yielding a topographic map. If the total area = 0, the still picture was excluded from the evaluation.

## Results

### Accuracy of training and validation

The accuracy rates for the training and validation datasets were 0.992 and 0.983, respectively. The accuracy rates for each identical class are listed in Table 3. In the training dataset, the accuracy for MES3 (0.951) was lower than that for the other categories and the validation dataset (0.952). Out of 21153 MES3 images for training, 940 images were mistrained, with inadequate quality for evaluation. The confusion matrix of training data after machine learning is shown in S1 Table. This was mainly because ulceration with exudate (Fig 4A and 4B) could not be discriminated from any residue covering the mucosal surface (Fig 4C and 4D). Table 4 shows the confusion matrix diagram indicating the results of classification using CNN. The true classes were on the vertical axis, and the predicted classes were on the horizontal axis. In the validation dataset, the number of MES3 images misclassified as MES0 was found to be larger (624) than the training dataset. These were mainly composed of images with minor ulcerations misclassified as white scars (Fig 4E and 4F). However, the accuracy rate of the validation data was greater than 0.98. Thus, this system could be used for automated severity evaluation of patients with UC undergoing CCE-2.

### Topography map of disease severity along the entire length of the colorectum

Fig 5 illustrates the topographic maps of severity in the same patient along the entire length of the colorectum (the cecum and ascending, transverse, descending and sigmoid, and rectum) before (Fig 5A) and after (Fig 5B) therapeutic intervention created by the still pictures obtained from a pair of forward (-f) and backward (-b) cameras. S2 and S3 Tables show the percentage of severity. The maps from the forward and backward cameras had an almost similar spatial distribution of disease severity. In this patient, the therapeutic intervention dramatically improved endoscopic disease severity and reduced disease extension, which has been correlated with clinical disease severity, including clinical activity index [27] (before 11 and after 2), fecal immunochemical test (before 2,155 ng/mL and after 1,037 ng/mL), and fecal calprotectin (before 14,100 μg/g and after 5,110 μg/g).

**Table 3. Accuracy of the training and test data set.** The accuracy for each category is presented next to the number of images.

| No | Categories | Training data set | | Validation data set | |
|---|---|---|---|---|---|
| | | correct images | accuracy | correct images | accuracy |
| 0 | MES0 | 112231 | 0.997 | 109067 | 0.994 |
| 1 | MES1 | 60107 | 0.996 | 13990 | 0.948 |
| 2 | MES2 | 8609 | 0.988 | 3203 | 0.913 |
| 3 | MES3 | 20107 | 0.951 | 19746 | 0.952 |
| 4 | Inadequate quality for evaluation | 278433 | 0.993 | 105046 | 0.986 |
| 5 | Ileal mucosa | 501 | 0.998 | 74 | 0.892 |
| | Total | 479988 | 0.992 | 251126 | 0.983 |

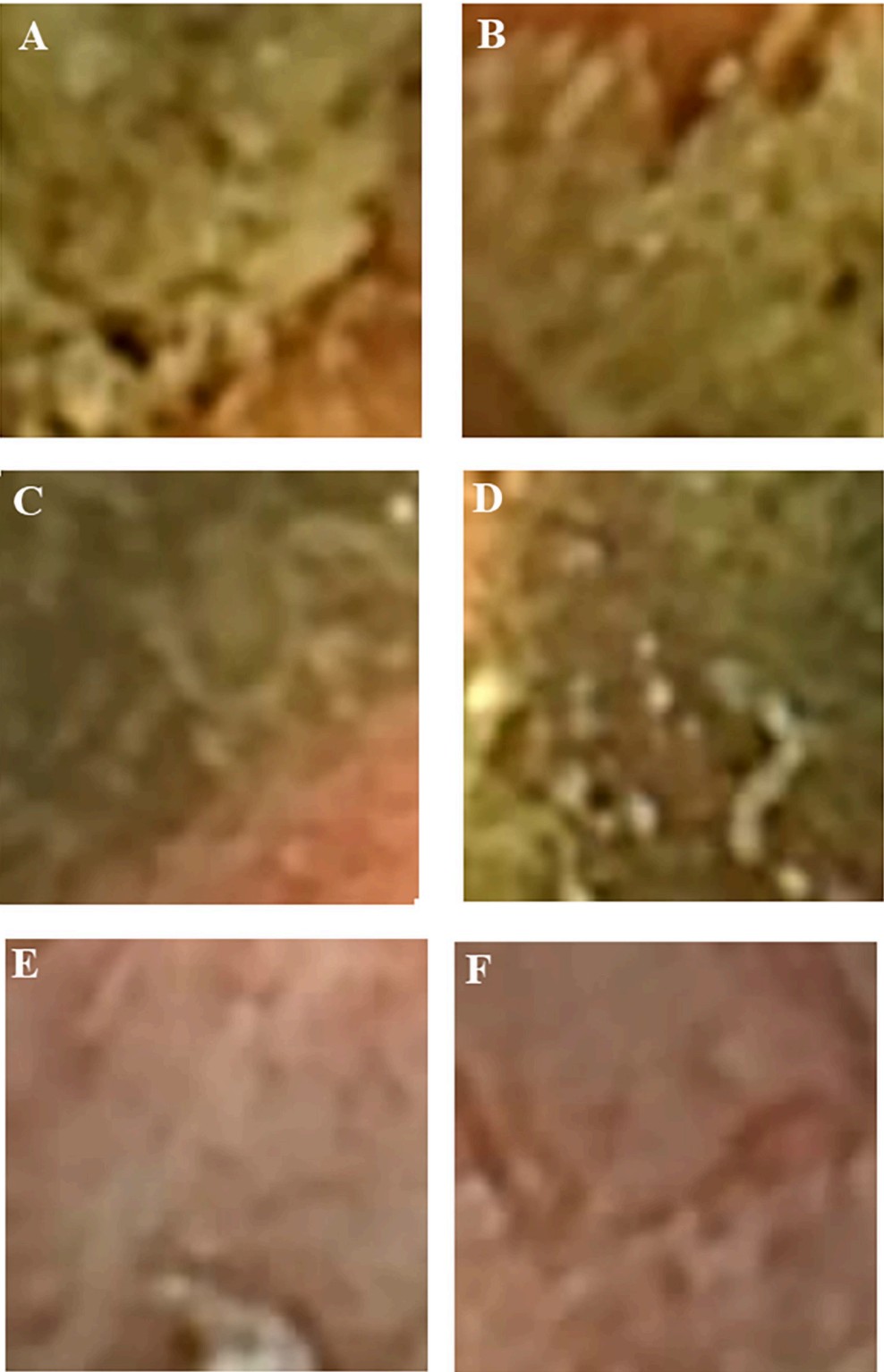

**Fig 4. Training images of ulceration.** A and B, examples for training images of ulceration with exudate labeled as MES3; C and D, examples for training images with opaque residue labeled as inadequate quality for evaluation which were not discriminated from A and B; E and F, examples for validation images with minor ulceration labeled as MES3, which were misclassified as white scar or MES0.

**Table 4. Confused matrix showing the classification results using established convolutional learning network.**

| true category / predicted category | | 0 | 1 | 2 | 3 | 4 | 5 |
|---|---|---|---|---|---|---|---|
| MES0 | 0 | 109067 | 315 | 3 | 8 | 256 | 72 |
| MES1 | 1 | 637 | 13990 | 12 | 23 | 87 | 4 |
| MES2 | 2 | 158 | 32 | 3203 | 41 | 73 | 0 |
| MES3 | 3 | 624 | 26 | 18 | 19746 | 322 | 0 |
| Inadequate quality for evaluation | 4 | 1345 | 129 | 14 | 43 | 105046 | 0 |
| Ileal mucosa | 5 | 9 | 0 | 0 | 0 | 0 | 74 |

## Discussion

In this study, we developed a disease severity classifier with high accuracy for capsule endoscopy video movies obtained from patients with UC. Dataset images with blackouts or higher-intensity areas were automatically excluded, and no other exclusion criteria for data cleansing by humans were set in a total of 731,114 training and validation datasets. This system provides clinicians with a topographic map of disease severity along the entire length of the colorectum in patients with UC, without requiring any invasive procedures. The process includes conversion of digest video files to serial still pictures, evaluation of topographic severity over a single still picture by ResNet50, yielding a stacked bar graph of % severity areas, and the synthesis of bar graphs to the topographic severity map in the colorectum.

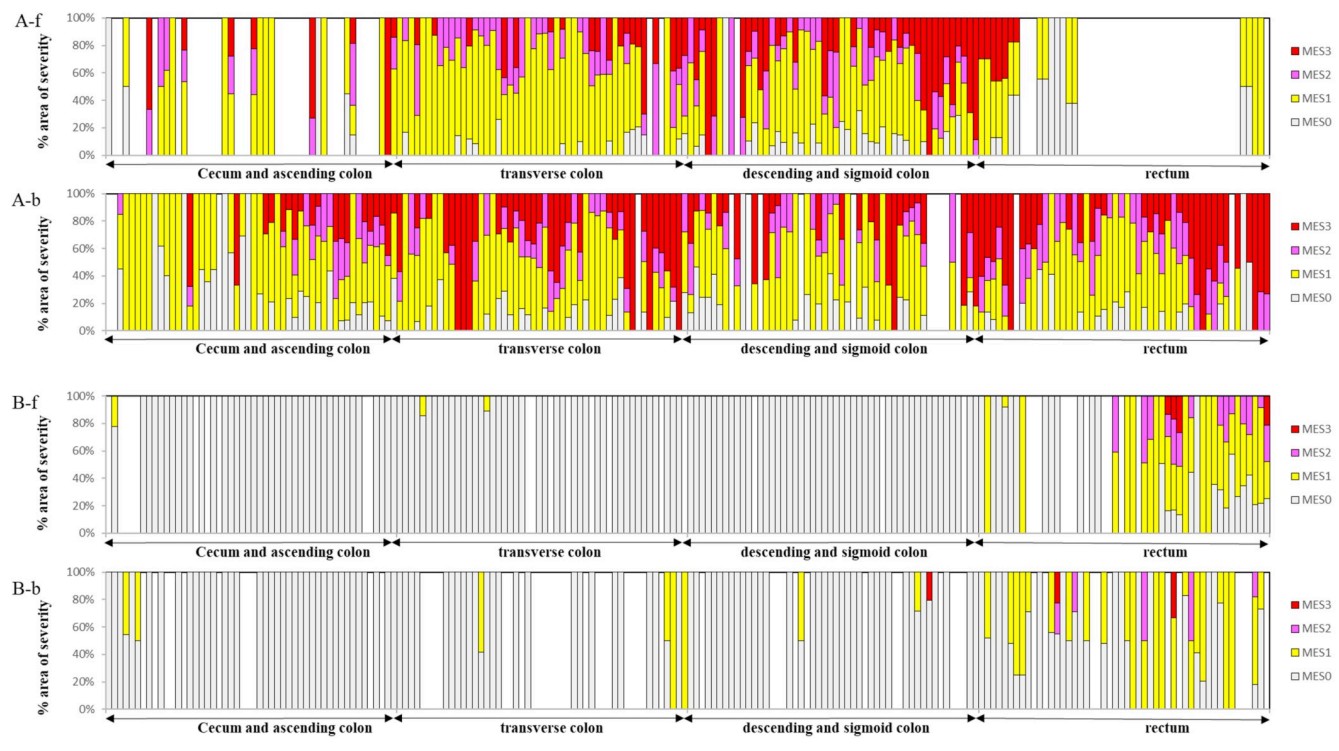

**Fig 5.** Examples of topography map of severity in the same patient along the entire length of the colorectum (the cecum and ascending, transverse, descending and sigmoid, and rectum) before (A) and after (B) therapeutic intervention. Suffix (-f) and (-b) indicate data files from forward and backward cameras, respectively. Severity is expressed by the stacked column composed of light gray (MES0%area), yellow (MES1%area), magenta (MES2%area), and red (MES3% area). The blank column corresponds to a still picture estimated as an inadequate condition for analysis.

The advancement of machine learning using CNN has enabled physicians to apply CAD to medical images in their various specialized fields. Stidham et al. [7] estimated the severity of UC using a GoogleNet-based CNN. Colonoscopy images of patients with UC were classified into two groups: the normal to mild group (Mayo score 0 or 1, and the moderate to severe group (Mayo 2 or 3). These metrics had an AUC of 0.966, sensitivity of 0.83, specificity of 0.96, positive predictive value of 0.87, and negative predictive value of 0.94. The authors constructed a 159-layer CNN to train and categorize images into two clinically relevant groups: remission (Mayo subscore 0 or 1) and moderate to severe disease (Mayo subscore, 2 or 3). The CNN was excellent for distinguishing endoscopic remission from moderate to severe disease, with an AUROC of 0.966 (95%CI, 0.967–0.972). Takenaka et al. [8] constructed the deep neural network for evaluating the UC (DNUC) algorithm. The DNUC identified patients with endoscopic remission with 90.1% accuracy (95% confidence interval [CI] 89.9%–90.9%). In addition, Ozawa et al. reported that a CNN-based CAD system was constructed based on GoogLeNet architecture [9]. The CNN-based CAD system showed a high level of performance, with AUROCs of 0.86 and 0.98 to identify Mayo 0 and 0–1, respectively.

Colonoscopy is the gold standard modality for evaluating pathology, disease severity, and its extension in patients with UC, but its low acceptability must be considered [11]. Wireless CCE was developed in 2006 to allow non-invasive visualization of the colon [13]. CCE-2 devices, such as PilCam COLON2, have enabled imaging that is superior to that of first-generation CCE. Therefore, CCE-2 is a potential modality for the routine assessment of disease severity and extension in patients with UC. However, drawbacks include the time-consuming and labor-intensive image interpretation, which sometimes takes more than an hour [28–30]. Junseok Park et al. [31] reported that the lesion detection assist using CNN significantly shortened the reading time of the capsule endoscope (1621.0–746.8 min for 20 videos; p = 0.029). To the best of our knowledge, this study is the first to develop a CNN-based CAD system for evaluating spatial disease severity along the entire length of the colorectum in routine CCE-2 examinations for patients with UC. The entire processing time was found to be accomplished within several minutes, with the burden of image interpretation by endoscopists subsequently reduced.

In this study, out of 739,021 patch images, 386,945 (52.4%) were classified as having inadequate quality for evaluation because of relatively poor bowel preparation. However, inflamed mucosa in patients with UC presents with diffuse extension, unlike neoplastic lesions, and the presence of at least one classified patch in a single still picture (Fig 3D) can allow the construct of a stacked bar graph, minimizing the number of still pictures excluded from evaluation.

To date, all studies on the endoscopic severity of patients with UC have assigned a single MES score to a single endoscopic image. However, MES may often vary depending on the location in a single picture (Fig 3A–3C), especially in the resolution phase [7–9, 32–36]. In addition, systematic disease severity along the length of the colorectum has been elusive because images are taken at undefined intervals during colonoscopy. The stacked bar chart composed of % severity area can evaluate the mixed MES scores, leading to the severity map along the entire colorectum, which may enable endoscopists to evaluate the effect of therapeutic interventions immediately and to decide on the appropriate therapeutic strategy (Fig 5).

The accuracy rate of MES3 in the training dataset (0.951) was lower than that of the other categories because ulceration with exudate could not be discriminated from residue covering the mucosal surface (Fig 4A and 4B). In the validation dataset, the accuracies of MES1 (0.948) and MES2 (0.913) were lower than those of the other categories. In misclassified images, a lower colonic air volume may obscure the interpretation of vascular patterns. The presence or absence of colonic vascularity has been evaluated precisely under sufficient colonic luminal air volume on colonoscopy. This is a limitation of capsule endoscopy when compared with

colonoscopy. Nevertheless, the first comparative study reported a high correlation (p = 0.797) between the severity evaluated using CCE-2 and colonoscopy [14].

Although the small parts (128 × 128 pixels) in the original still picture were used for training or validation datasets, even an expert endoscopist could not correctly classify similar-looking small images without reference to the whole still picture series, which may lead to misclassification between the above-stated ulcer base and residue issue. However, small images, as in this dataset, are required to evaluate topographic severity. In this context, diagnostic imaging must be conducted to address scaling problems. Nonetheless, the constructed CNN offered good performance for fully automated severity mapping in patients with UC and could reduce the burden on clinicians, including experts and non-expert endoscopists.

## Limitations

This study had several limitations. First, this was a single-center study. The use of data with a variety of severities or extensions from other institutions might improve degradation performance and prevent overfitting. Second, classification criteria for disease severity were established based on colonoscopic findings, which precluded the precise application of some factors such as vascularity. Third, liquid preparation could have caused flare-ups among the patients with severe lesions. Thus, we had excluded UC patients with severe conditions—as determined by the questionnaire—from this study. Consequently, generalizability of the results of this study may have been impaired because of this selection bias. Fourth, in this study, a large number of images (52.4%) were classified as "inadequate quality for analysis"; this was considered to be caused by the preparation regimen that was not necessarily optimized for this study. However, we do not think that this large number of unanalyzed images could have introduced any type of bias thereby giving us imprecise results, becauseUC lesions were rarely missed by capsule endoscopy owing to their diffuse distribution. Although there is plenty of scope for improvement in the pre-treatment regimen, this diffuse distribution of UC could have nullified the possible bias of poor-quality images. Finally, small trimmed patch images could not be evaluated among those obtained by capsule endoscopy, therefore, our diagnoses did not include UC-complicated intestinal lesions, such as cytomegalovirus enteritis and UCAN.

## Conclusion

The created disease severity classifier for patients with UC enabled fully automated severity mapping on capsule endoscopy. This system may reduce the burden on endoscopists regarding time-consuming image interpretation for therapeutic outcomes and may be developed into a standard severity evaluation tool for an optimized therapeutic regimen.

## Supporting information

**S1 Table. This is confused matrix showing the classification results of training data using established convolutional neural network.** S1 Table shows the confusion matrix diagram of training data indicating the results of classification using CNN. The true classes were on the vertical axis, and the predicted classes were on the horizontal axis.
(XLSX)

**S2 Table. This is percentage of severity of Fig 5.** S2 Table shows percentage of severity in forward and backward cameras before changing the therapeutic intervention.
(XLSX)

**S3 Table. This is percentage of severity of Fig 5.** S3 Table shows after the therapeutic intervention. In these tables, the severity classification and proportion of each still image are shown

on the horizontal axis.
(XLSX)

## Author Contributions

**Conceptualization:** Naoki Higuchi, Hiroto Hiraga, Yoshihiro Sasaki, Shohei Igarashi, Keisuke Hasui, Yasuhisa Murai, Tetsuya Tatsuta, Hidezumi Kikuchi, Daisuke Chinda, Tatsuya Mikami, Hirotake Sakuraba, Shinsaku Fukuda.

**Data curation:** Naoki Higuchi, Hiroto Hiraga.

**Formal analysis:** Naoki Higuchi, Hiroto Hiraga, Masashi Matsuzaka.

**Investigation:** Naoki Higuchi, Hiroto Hiraga, Noriko Hiraga, Keisuke Hasui, Kohei Ogasawara, Takato Maeda.

**Methodology:** Naoki Higuchi, Hiroto Hiraga, Yoshihiro Sasaki, Shohei Igarashi, Kohei Ogasawara, Hidezumi Kikuchi.

**Project administration:** Masashi Matsuzaka, Hirotake Sakuraba.

**Resources:** Noriko Hiraga, Keisuke Hasui, Takato Maeda, Hirotake Sakuraba.

**Software:** Yoshihiro Sasaki.

**Supervision:** Naoki Higuchi, Yoshihiro Sasaki, Tatsuya Mikami, Hirotake Sakuraba, Shinsaku Fukuda.

**Validation:** Naoki Higuchi, Yoshihiro Sasaki, Shohei Igarashi.

**Visualization:** Naoki Higuchi, Hiroto Hiraga, Yoshihiro Sasaki, Tetsuya Tatsuta.

**Writing – original draft:** Naoki Higuchi, Hiroto Hiraga, Yoshihiro Sasaki.

**Writing – review & editing:** Naoki Higuchi.

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
