## [Decision Letter · Decision Letter 0]

24 Feb 2022

PONE-D-22-01439Automated evaluation of colon capsule endoscopic severity of ulcerative colitis using ResNet50PLOS ONE

Dear Dr. Hiraga,

Thank you for submitting your manuscript to PLOS ONE. After careful consideration, we feel that it has merit but does not fully meet PLOS ONE’s publication criteria as it currently stands. Therefore, we invite you to submit a revised version of the manuscript that addresses the points raised during the review process.

Your manuscript was assessed by two expert reviewers, and substantive issues have been raised as listed below. Regarding the criticisms from both reviewers, you don't need to perform additional experiments but must fully address the issues in the revised results and discussion sections.

We look forward to receiving your revised manuscript.

Kind regards,

Emiko Mizoguchi, M.D., Ph.D.

Academic Editor

PLOS ONE

Journal Requirements:

2. PLOS requires an ORCID iD for the corresponding author in Editorial Manager on papers submitted after December 6th, 2016. Please ensure that you have an ORCID iD and that it is validated in Editorial Manager. To do this, go to ‘Update my Information’ (in the upper left-hand corner of the main menu), and click on the Fetch/Validate link next to the ORCID field. This will take you to the ORCID site and allow you to create a new iD or authenticate a pre-existing iD in Editorial Manager. Please see the following video for instructions on linking an ORCID iD to your Editorial Manager account: https://www.youtube.com/watch?v=_xcclfuvtxQ.

4.  We noticed you have some minor occurrence of overlapping text with the following previous publication(s), which needs to be addressed:

- https://www.sciencedirect.com/science/article/abs/pii/S0010482520302857?via%3Dihub

In your revision ensure you cite all your sources (including your own works), and quote or rephrase any duplicated text outside the methods section. Further consideration is dependent on these concerns being addressed.

Reviewers' comments:

Reviewer's Responses to Questions

**Comments to the Author**

1. Is the manuscript technically sound, and do the data support the conclusions?

Reviewer #1: Yes

Reviewer #2: Yes

2. Has the statistical analysis been performed appropriately and rigorously? 

Reviewer #1: Yes

Reviewer #2: Yes

3. Have the authors made all data underlying the findings in their manuscript fully available?

Reviewer #1: Yes

Reviewer #2: Yes

4. Is the manuscript presented in an intelligible fashion and written in standard English?

Reviewer #1: Yes

Reviewer #2: Yes

5. Review Comments to the Author

Reviewer #1: This is an exciting innovation for computer-aided diagnosis for UC. I have two questions.

1. As authors stated, the large number of images (52.4%) have been classified as "inadequate quality for analysis". Do authors think that this examination is intrinsically reliable and reproducible? Please describe about it.

2. How much amount of liquid preparation did the patients have in MOVIPREP method for colon capsule? Is it tolerable and ethically acceptable for UC patients with severe MES 3? In severe condition of UC, most clinicians evaluate the severity by sigmoidoscopy without oral liquid preparation which may cause flare-up. Please describe about it.

Reviewer #2: This paper presents a very interesting study on the development of an automated severity assessment system (ResNet50) that can evaluate the entire colon of patients with ulcerative colitis using modern colon capsule endoscopy.

What is most impressive is that the activity of the entire colon can be easily monitored in a bird's eye view, as shown in Figure 5. However, although this is not the main purpose of this study, there may be differences in opinion among institutions as to whether CCE with a higher amount of premedication is better tolerated than colonoscopy. This technology may be applicable to regular colonoscopy in the future.

I have several concerns that need to be addressed before considering publication.

① Looking at the images in Figure 1-4, I have the impression that bleeding is not sufficiently evaluated. Bleeding is very important in assessing the severity of the disease, but how is it assessed in this study? Please add this to your manuscript on the evaluation method.

② Also related to the above, as shown in existing related reports1 2 , Compared to MES, UCEIS is generally considered to be a more accurate assessment of mucosal activity. What was the reason you did not evaluate it in the UCEIS? Also, if you believe that MES is sufficient for evaluation, please explain the reason.

③ For Figure 5, I think it would be easier to visualize the activity of the entire colon if there were 3D graphs. If possible, please considering add those images.

④ How are UC-complicated intestinal lesions, such as cytomegalovirus enteritis and UCAN, diagnosed? Or are they excluded? Please add this to your manuscript.

1. Hosoe N, Nakano M, Takeuchi K, et al. Establishment of a Novel Scoring System for Colon Capsule Endoscopy to Assess the Severity of Ulcerative Colitis-Capsule Scoring of Ulcerative Colitis. Inflamm Bowel Dis 2018;24:2641-2647.

2. Takano R, Osawa S, Uotani T, et al. Evaluating mucosal healing using colon capsule endoscopy predicts outcome in patients with ulcerative colitis in clinical remission. World J Clin Cases 2018;6:952-960.

6. PLOS authors have the option to publish the peer review history of their article (what does this mean?). If published, this will include your full peer review and any attached files.

Reviewer #1: No

Reviewer #2: No

---

## [Author Response · Author response to Decision Letter 0]

18 Apr 2022

Dear Dr. Chenette, Dr.Mizoguchi:

Thank you for giving me the opportunity to submit a revised draft of my manuscript titled “Automated evaluation of colon capsule endoscopic severity of ulcerative colitis using ResNet50” with PLOS ONE. We appreciate the time and effort that you and the reviewers have dedicated to providing your valuable feedback on my manuscript. We are grateful to the reviewers for their insightful comments on our paper. We have been able to incorporate changes to reflect most of the suggestions provided by the reviewers. We have indicated the changes within the manuscript in red, light blue, and green font colors. Here is a point-by-point response to the reviewers’ comments and concerns. 

・Response to journal requirements: Thank you for your valid comments. I agree with all four of the suggestions and have incorporated them to the revised version of the manuscript (green color). 

1. Yes

2. Yes

3. Yes

4. I agree with you. I have inserted the following citation in the revised manuscript.

-https://www.sciencedirect.com/science/article/abs/pii/S0010482520302857?via%3Dihub

 Lastly, according to the changes mentioned above, I have deleted some redundant sentences.

The deleted sentences were on Page 3, Lines 54-55 of the original version of the manuscript.

・Response to Reviewer #1: Thank you for giving me good advice. I agree with both the suggestions and have revised the manuscript accordingly (changes are indicated in red font color). 

1. As authors stated, the large number of images (52.4%) have been classified as "inadequate quality for analysis". Do authors think that this examination is intrinsically reliable and reproducible? Please describe about it.

Response: There were images that were taken randomly along the lesions that were of poor quality. We do not think that the large number of unanalyzed images, because of inadequate quality, introduced any kind of bias that resulted in imprecise results in this study. The UC lesions were missed rarely by capsule endoscopy, because of their diffuse distributions. This characteristic of the disease has potentially contributed to avoiding the bias. We have incorporated this point in the limitations of the revised manuscript (lines 312–318 of the revised manuscript).

2. How much amount of liquid preparation did the patients have in MOVIPREP method for colon capsule? Is it tolerable and ethically acceptable for UC patients with severe MES 3? In severe condition of UC, most clinicians evaluate the severity by sigmoidoscopy without oral liquid preparation which may cause flare-up. Please describe about it.

Response: We excluded UC patients with severe conditions—as determined by the questionnaire—from this study. Additionally, it is not possible to predict very severe lesions in patients without severe subjective symptoms prior to investigations. No severe subjective symptoms developed in the participants with MES 3 at the time of the examination. Nevertheless, we know that if UC patients with severe subjective symptoms had undergone the MOVIPREP regimen, it would have been intolerable and unacceptable to them. Based on the aforementioned reasons, we think that the methods of the study were tolerable and ethically correct for the patients. Although, we do acknowledge that liquid preparation could have caused flare-ups among the patients with severe lesions, as the reviewer pointed out. Accordingly, we have added this limitation to the revised manuscript (lines 308–312 of the revised manuscript).

I believe that incorporating your advice into the revised version has made the manuscript better. Thank you once again.

・Response to Reviewer #2: Thank you for the excellent advice. I agree with all four of your suggestions and have revised the manuscript accordingly (revisions are indicated in light blue color font). 

1. Bleeding is very important in assessing the severity of the disease, but how is it assessed in this study? Please add this to your manuscript on the evaluation method.

Response: I agree with your suggestion. Bleeding is very important in assessing the severity of the disease, especially in patients with severe UC. However, we excluded UC patients with severe conditions from this study, because MOVIPREP method would have been intolerable and it would have been ethically wrong to cause harm. We have added the exclusion criterion aforementioned in the method section of the revised manuscript (lines 134–136 of the revised manuscript).

2. What was the reason you did not evaluate it in the UCEIS? Also, if you believe that MES is sufficient for evaluation, please explain the reason.

Response: Although UCEIS is recognized to be a more accurate assessment of mucosal severity in patients with UC as compared to MES, UCEIS is designed to evaluate the severity with a single image. In contrast, the advantage of capsule endoscopy is to obtain and evaluate serial images of the whole colon. Moreover, our procedure can construct a topographic map of the severity. To augment these assets, we selected MES and not UCEIS to evaluate the severity. We have added the reasons for adopting MES in the method section of the revised manuscript (lines 160–165 of the revised manuscript). 

3. For Figure 5, I think it would be easier to visualize the activity of the entire colon if there were 3D graphs. If possible, please considering add those images.

Response: We converted Figure 5, which was composed of two-dimension graphs, into three-dimension graphs, but, unfortunately, they were not easy to understand and did not add new information for readers. If the reviewer can please suggest to us what is required more specifically, we will try put it into practice.

4. How are UC-complicated intestinal lesions, such as cytomegalovirus enteritis and UCAN, diagnosed? Or are they excluded? Please add this to your manuscript.

Response: Small trimmed patch images could not be evaluated among those obtained by capsule endoscopy, so our diagnoses did not include UC-complicated intestinal lesions, such as cytomegalovirus enteritis and UCAN. We recognized it as a limitation of this study, and have added it to the limitation subsection of the revised manuscript (lines 318–321 of the revised manuscript).

I believe that incorporating your advice in the revised version has made the manuscript better. Thank you once again.

---

## [Decision Letter · Decision Letter 1]

27 May 2022

Automated evaluation of colon capsule endoscopic severity of ulcerative colitis using ResNet50

PONE-D-22-01439R1

Dear Dr. Hiraga,

We’re pleased to inform you that your manuscript has been judged scientifically suitable for publication and will be formally accepted for publication once it meets all outstanding technical requirements.

Kind regards,

Emiko Mizoguchi, M.D., Ph.D.

Academic Editor

PLOS ONE

Additional Editor Comments (optional):

Reviewers' comments:

Reviewer's Responses to Questions

**Comments to the Author**

1. If the authors have adequately addressed your comments raised in a previous round of review and you feel that this manuscript is now acceptable for publication, you may indicate that here to bypass the “Comments to the Author” section, enter your conflict of interest statement in the “Confidential to Editor” section, and submit your "Accept" recommendation.

Reviewer #1: All comments have been addressed

Reviewer #2: All comments have been addressed

2. Is the manuscript technically sound, and do the data support the conclusions?

Reviewer #1: Yes

Reviewer #2: Yes

3. Has the statistical analysis been performed appropriately and rigorously? 

Reviewer #1: Yes

Reviewer #2: Yes

4. Have the authors made all data underlying the findings in their manuscript fully available?

Reviewer #1: Yes

Reviewer #2: Yes

5. Is the manuscript presented in an intelligible fashion and written in standard English?

Reviewer #1: Yes

Reviewer #2: Yes

6. Review Comments to the Author

Reviewer #1: (No Response)

Reviewer #2: I have received sufficient answers to my questions and suggestions.

Please check one point. There does not appear to be a full spelling of UCAN in the text. Please check and add it.

7. PLOS authors have the option to publish the peer review history of their article (what does this mean?). If published, this will include your full peer review and any attached files.

Reviewer #1: No

Reviewer #2: No

---

## [Editor Report · Acceptance letter]

2 Jun 2022

PONE-D-22-01439R1 

Automated evaluation of colon capsule endoscopic severity of ulcerative colitis using ResNet50 

Dear Dr. Hiraga:

I'm pleased to inform you that your manuscript has been deemed suitable for publication in PLOS ONE. Congratulations! Your manuscript is now with our production department. 

Kind regards, 

on behalf of

Dr. Emiko Mizoguchi 

Academic Editor

PLOS ONE